# Does Preoperative Cognitive Optimization Improve Postoperative Outcomes in the Elderly?

**DOI:** 10.3390/jcm11020445

**Published:** 2022-01-15

**Authors:** Yumiko Ishizawa

**Affiliations:** Department of Anesthesia, Critical Care and Pain Medicine, Massachusetts General Hospital, Harvard Medical School, Boston, MA 02114, USA; yishizawa@mgh.harvard.edu; Tel.: +1-617-726-3030

**Keywords:** preoperative optimization, cognitive training, perioperative neurocognitive disorder, postoperative delirium, advanced age

## Abstract

Perioperative neurocognitive disorder (PND) is a growing concern, affecting several million elderly patients each year in the United States, but strategies for its effective prevention have not yet been established. Humeidan et al. recently demonstrated that preoperative brain exercise resulted in a decrease in postoperative delirium incidence in elderly surgical patients, suggesting the potential of presurgical cognitive optimization to improve postoperative cognitive outcomes. This brief review summarizes the current knowledge regarding preoperative cognitive optimization and highlights landmark studies, as well as current ongoing studies, as the field is rapidly growing. This review further discusses the benefit of cognitive training in non-surgical elderly populations and the role of cognitive training in patients with preexisting cognitive impairment or dementia. The review also examines preclinical evidence in support of cognitive training, which can facilitate understanding of brain plasticity and the pathophysiology of PND. The literature suggests positive impacts of presurgical cognitive optimization, but further studies are encouraged to establish effective cognitive training programs for elderly presurgical patients.

## 1. Introduction

Perioperative neurocognitive disorder (PND) affects a growing population of older adults. Dementia after general anesthesia in older patients was first reported in 1955 [1]. However, the reversibility of changes in consciousness and cognition after anesthesia and surgery had not been seriously studied until a few decades ago. Since the first large clinical trial was published in 1998, in which it was suggested that age is a major risk factor for late cognitive dysfunction after anesthesia and surgery [2], postoperative cognitive dysfunction or PND is now widely recognized as a disorder that affects patients undergoing surgery, especially patients of advanced age. PND, including postoperative delirium (POD), is known to be associated with an overall increase in morbidity and mortality [3,4,5]. Over 16 million patients aged 65 and older undergo anesthesia and surgery each year in the United States [6]. Studies have suggested that 5–54% of these patients will develop a form of PND up to 1 year after surgery [7,8]. Thus between 0.8 and 9 million patients will likely suffer from PND each year in the US [7]. The population of older adults is rapidly growing, both in the US and worldwide, and the population having surgery is aging at a faster rate than the general population [9]. Furthermore, patients with PND experience a much higher incidence of new disability after surgery [10] and are at increased risk of leaving the labor market prematurely and becoming dependent on social transfers [5]. A recent cohort study of Medicare patients aged 65 years or older suggests that a diagnosis of PND was associated with an increase in healthcare costs for up to 1 year following surgery [11]. Preventing PND will significantly extend healthier human lifespan and reduce healthcare costs [12,13].

Despite the critical importance of this issue in our aging society, the pathophysiology of PND is poorly understood. Effective prevention and treatment have not been established [4,5,14,15,16]. PND is consistently and predominantly observed as deficits in memory and executive function [17,18]. Older age and preexisting cognitive impairment are the most important preoperative risk factors (predisposing factors) for PND [19,20]. Precipitating factors such as anesthetic drugs, surgery, and pain are thought to trigger POD or to promote cognitive dysfunction [21,22]. Multiple theories have been proposed for the pathophysiology of PND. These include neuroinflammation, neuronal injury, neurotransmitter abnormalities, and the unmasking or acceleration of preexisting neurocognitive disorders [17,23,24]. No successful clinical applications have yet been shown based on the possible pathophysiology. On the other hand, efforts have been made to optimize predisposing factors of PND, such as preexisting cognitive impairment. This brief review aims to summarize the current knowledge regarding preoperative optimization, focusing on cognitive optimization, and will discuss whether preoperative optimization improves postoperative cognitive outcomes in elderly surgical patients. A focused literature search for articles published between the years 2000 and 2021 was performed in PubMed using relevant terms, including preoperative optimization, cognitive training, perioperative neurocognitive disorder, postoperative delirium, and advanced age. Past and current clinical trials were also searched at ClinicalTrials.gov. The review highlights landmark studies in the field and discusses ongoing studies, as the field is rapidly growing. In addition to emerging clinical findings, preclinical evidence for cognitive optimization is briefly discussed. This may help in understanding the pathophysiology of PND and guide future investigations on the role of cognitive optimization in preventing PND and age-related cognitive disorders.

## 2. Preoperative Cognitive Assessment

**Presurgical cognitive impairment and dementia.** Preexisting cognitive impairment has been identified as one of the major risk factors for PND [3,15,17]. The prevalence of presurgical cognitive impairment or dementia appears to vary in different patient groups [25], and a wide range of elderly presurgical patients (19–83%) have been reported to have preexisting cognitive impairment [26]. During the preoperative period, a thorough assessment of medical and social history should be made. The assessment should include the patient’s decision-making capacity; any history of depression and/or other psychiatric illnesses; their functional dependence, frailty, nutritional status, alcohol and drug use; and the patient’s list of medications [26]. Thus, predisposing and precipitating factors of PND can be identified and modifiable risk factors may be optimized prior to surgery. Comprehensive geriatric assessment is an approach involving multi-domain assessment and optimization [27,28], and can be a useful tool during the preoperative period to identify potentially modifiable risk factors in elderly surgical patients. Studies suggest that comprehensive geriatric assessment has a positive impact on postoperative outcomes [29,30,31]. Medically treatable conditions that may contribute to cognitive impairment include vitamin deficiencies, metabolic abnormalities, medication effects, and intracranial fluid accumulation (i.e., normal pressure hydrocephalus). These conditions should be diagnosed and treated prior to surgery. Thus, comprehensive geriatric assessment is ideal for elderly surgical patients. Current multidisciplinary guidelines emphasize cognitive assessment and recommend preoperative cognitive screening of older adults undergoing surgery [7,17,26,32,33,34]. However, the recommendations are infrequently met in many institutions. Cognitive screening and optimization require a multidisciplinary team, advanced planning, and sufficient preoperative time. The lack of any of these elements is a barrier for implementing a cognitive screening and optimization program. Further, there are multiple standardized tools used for cognitive assessment, but the tools chosen vary in different programs [26].

Most preoperative assessment programs are currently provided remotely, especially since the COVID-19 pandemic. However, a cognitive assessment that can be performed entirely remotely by phone or video call is not available and has not been tested for clinical use. Recent studies investigated the effectiveness of brief cognitive testing in emergency or preoperative settings. Ruiz et al. proposed a brief physical and cognitive assessment that can be performed in an emergency setting, and demonstrated that preexisting physical and cognitive deficits contribute to postoperative complications in the elderly patients [35]. Tiwary et al. studied the use of the Mini-Cog test immediately before surgery in elderly surgical patients and demonstrated that the Mini-Cog results on surgery day and during the preoperative clinic visit (average 8.4 days before surgery) showed high agreement [36]. They also reported that both scores were highly predictive of POD in the recovery room.

Importantly, elderly patients are often undiagnosed with cognitive impairment or dementia until they develop POD or PND [26]. Although the exact prevalence of undiagnosed dementia in elderly surgical patients is unknown, Alzheimer’s Disease International reported that over 55 million people lived with dementia worldwide in 2019 and that globally, 75% of people with dementia are not diagnosed [37]. A retrospective cohort study reported cognitive deficits in 23.5% of the unselected elderly surgical patients aged 65 years and older during preoperative cognitive screening. Their deficits were associated with increased age, decreased function, decreased BMI, and several common medical comorbidities [38]. A prospective cohort study from South Africa using the Mini-Cog test reported undiagnosed cognitive impairment in 57% of older surgical patients (median age: 65 years old) [39]. Low scores were associated with increased age, unskilled occupation, low level of education, low functional status, and frailty. These numbers suggest that anesthesiologists will be confronted with the management of increasing numbers of patients with undiagnosed cognitive impairment. It is thus imperative to have standardized diagnostic tools that can be used in various preoperative settings to identify undiagnosed cognitive impairment, so that preventative efforts for PND can be initiated.

**Frailty syndrome and cognitive impairment.** Frailty is a clinical syndrome, the importance of which has been rapidly recognized in perioperative elderly populations [40]. Frailty syndrome is measured as the sum of various indicators, including unintentional weight loss, fatigue, muscle weakness, low physical activity, low gait speed, poor balance, visual impairment, and cognitive impairment [41,42]. The prevalence of preoperative frailty varies significantly (up to 50–60%) depending on age and the type of surgery, with a steep increase observed in older patients [43,44,45,46]. Preoperative frailty is known to be associated with poor postoperative morbidity and mortality [47]. Whether frailty is associated with PND is being actively investigated. Gracie et al. recently conducted a meta-analysis of nine qualified clinical studies and demonstrated evidence of an association between preoperative frailty and POD in elective surgical patients aged 65 years or older when compared to nonfrail control patients [43]. The association of frailty with POD has also been suggested in multiple other recent studies [48,49,50,51,52]. However, it remains unclear whether any particular components of frailty syndrome drive the association with POD, or if preoperative cognitive impairment by a single entity is a major factor driving this association. In a prospective cohort study, Susano et al. reported that preoperative frailty and cognitive impairment were independently associated with POD [44]. Whether preoperative optimization improves postoperative cognitive outcomes in patients with frailty syndrome needs further investigation [53].

## 3. Presurgical Cognitive Optimization

**Prehabilitation.** Prehabilitation is a wide-spread concept aiming to enhance general health or optimize comorbidities prior to major surgery [54,55,56]. Prehabilitation originally focused on the improvement of physical ability and nutritional status but is moving towards a multimodal approach that includes stress and anxiety reduction and mind-body prehabilitation. Psychological factors are increasingly recognized as an important element of prehabilitation, and are often added to prehabilitation programs (trimodal or multimodal prehabilitation) [54]. Fulop et al. reported that a trimodal prehabilitation program, in which adult patients underwent 3–6 weeks of physical, emotional, and nutritional treatments prior to colorectal surgery, improved functional status and some parameters of emotional and physical well-being, but did not change postoperative morbidity or mortality [57]. In older surgical patients, Liu et al. conducted a meta-analysis which demonstrated that trimodal prehabilitation improved postoperative functional status but did not reduce postoperative mortality and complications [58]. A multicenter, prospective, randomized controlled trial investigating the effect of multimodal prehabilitation on functional capacity and postoperative complications in adult colorectal surgical patients is currently underway [59]. However, these studies do not include cognitive prehabilitation or cognitive outcome measurements, which still require future investigation.

**Effects of cognitive training in older non-surgical populations.** Before any discussion of the effect of cognitive training in surgical patients, it is important to know how cognitive training can contribute to the improvement of cognitive function in non-surgical patients. Normal aging is associated with progressive functional losses in perception, cognition, and memory [60]. However, brain plasticity, which refers to the brain’s lifelong capacity for physical and functional changes, has been shown in the aged brain [60], suggesting that the aged brain can be optimized presurgically. There is increasing evidence that cognitive interventions improve targeted cognitive abilities in older adults [60,61]. Limited studies further demonstrate the generalizability of these effects to untrained cognitive abilities and prolonged maintenance of improvements [60]. A pioneer study by Mahncke et al. demonstrated that brain-plasticity-based intensive training (aural language reception accuracy) improved trained auditory and language functions and nontrained memory tasks [60]. The improvement of those functions was sustained after the completion of training. Moreover, the Advanced Cognitive Training for Independent and Vital Elderly (ACTIVE) study demonstrated that three distinct cognitive training interventions (memory, reasoning, speed of processing) in ten 60–75 min sessions effectively improved targeted cognitive abilities in adults aged 65 and older [61]. At the 10-year follow up, cognitive training for reasoning and speed of processing, but not for memory, showed beneficial effects on trained cognitive abilities. The study participants self-reported less decline in daily functioning compared with the control group [62]. A meta-analysis of seven randomized controlled studies of cognitive intervention in the elderly suggests long-lasting protective effects on cognition in healthy older adults for up to 6 years [63]. Whether cognitive interventions can prevent incident dementia has not been shown [63].

Anguera et al. investigated the underlying neurophysiology of the effects of cognitive training using a custom-designed video game [64]. They demonstrated that older adults (60–85 years old) who completed the video game training in multitasking mode (signal detection and car control on the game) for a total of 12 h over 4 weeks showed improved multitasking performance for 6 months. Interestingly, only the multitasking training group demonstrated improvements in untrained cognitive control tasks such as working memory and sustained attention, indicating the transfer of benefits to the cognitive abilities that are known to be most impaired by aging. Electroencephalography (EEG) during the video game performed before and after the 4 week training showed for the first time that multitasking training induced increases in midline frontal theta power—a change which positively correlated with both sustained multitasking performance improvements and sustained attention. Six years later, the multitasking training participants continued to show improvements, with a neural signature of cognitive control, that is, midline frontal theta power [65]. However, the previously evidenced transfer of benefits extended to untrained cognitive abilities (i.e., enhanced working memory) was not persistent. These data together suggest limited but robust plasticity of the cognitive control system in the aged brain. These studies reveal both neurophysiological and behavioral evidence of the long-term effects of cognitive training in older adults. Whether these EEG changes are seen with other types of cognitive training is yet to be determined. The EEG, however, has the potential to be used as a training monitor to individually guide the optimal dose of cognitive training in non-surgical and surgical populations.

Importantly, there are studies which have investigated the effect of cognitive training in non-surgical adults with cognitive impairment. A meta-analysis by Hill et al. showed that computerized cognitive training improved cognitive functions in older adults with mild cognitive impairment [66]. The effect size was larger than the effect seen in healthy older adults [67]. A recent study suggests that traditional paper-based cognitive training improved cognitive functions in multiple domains in patients with early-stage Alzheimer’s disease [68]. However, evidence is still limited for the effects of cognitive training in older adults with Alzheimer’s disease and other types of dementia [66,69].

**Preoperative cognitive training and postoperative outcomes.** Cognitive optimization is a relatively new concept compared to presurgical optimization for other specific organ systems. In fact, optimizing congestive heart failure, hypertension, diabetes, or kidney failure is a core practice of presurgical assessment with positive outcomes [70,71,72]. Whether presurgical cognitive optimization can improve cognitive outcomes in elderly patients is a critical question, considering the growing importance of PND and its potential association with Alzheimer’s disease and related dementias [14,73]. Presurgical cognitive training has the potential to be implemented swiftly in various settings since cognitive training is thought to be without serious consequences. Humeidan et al. recently demonstrated that preoperative brain exercise resulted in a decrease in delirium incidence in noncardiac and nonneurological surgical patients aged 60 years and older [74]. This randomized single-blinded clinical trial used a tablet-based cognitive exercise targeting memory, speed, attention, flexibility, and problem-solving functions. The delirium rate was significantly lower in the cognitive exercise group compared to the control participants (13.2% vs. 23.0%, *p* = 0.04), but there were no differences in the onset day, duration, or total delirium-positive days between study groups. The data are promising, but the effect on long-term cognitive dysfunction or overall PND requires further research. In addition, the ideal type of cognitive training, timing, and dosage will be valuable information for extending the clinical applications of presurgical cognitive optimization.

A number of feasibility studies and clinical trials are currently under way to further examine the effect of presurgical cognitive training. The feasibility of perioperative cognitive training via a mobile device was demonstrated in a study of older adults (60–90 years old) undergoing cardiac surgery [75]. The study did not show significant differences in the incidence of postoperative delirium and cognitive dysfunction between the cognitive training group and usual care group, but cognitive training participants agreed that their memory and thinking ability had improved. On the other hand, short-term, home-based, unsupervised preoperative cognitive training using a computer-based cognitive battery is unlikely to be feasible for patients aged 60 years or older undergoing noncardiac surgery [76]. The barriers to training appear to include feeling overwhelmed, technical difficulties, and preoperative time commitment. The authors pointed out that enrollment in the program over a short timeframe may worsen preoperative anxiety and apprehension [76]. Computer-based cognitive training provided both preoperatively and postoperatively is currently being studied in older patients undergoing coronary artery bypass grafting surgery [77], where the cognitive domains most affected by heart failure (e.g., psychomotor speed, attention, memory, and executive function) are under investigation. Further, a randomized pilot study combining preoperative cognitive training and physical exercise prehabilitation is underway in older surgical patients [78].

In addition, there are clinical trials being conducted to evaluate a multidisciplinary, multicomponent perioperative intervention that does not include a specific cognitive training program [79,80]. The Perioperative Enhancement of Cognitive Trajectory (PROTECT) trial is a randomized, controlled, multicenter, single-blind clinical trial that investigates a multidisciplinary perioperative intervention. It includes preoperative psychoeducation and delirium risk reduction strategies such as nutrition, hydration, exercise, and stress management recommendations [79]. This study will provide valuable information on how general preoperative optimization without cognitive training improves cognitive trajectory. The feasibility of a multicomponent preoperative and intraoperative intervention including comprehensive preoperative counselling and sensory orientation aids is also being investigated for a variety of surgeries, with the minimum exclusion of neurocerebral and ophthalmological surgeries [80].

## 4. Preclinical Studies

Comprehensive behavioral and neurophysiological studies in preclinical animal models are essential to understand the effect of preoperative interventions on the pathophysiology of PND [16]. Preoperative exercise and enriched environment have been shown to attenuate postoperative cognitive impairment in multiple aged rodent studies [81,82,83,84]. Both exercise and enriched environment appear to prevent neuroinflammation in these animal models [82,84]. However, the effect of cognitive optimization or cognitive training on PND has not been reported in aged animal models.

On the other hand, studies on nonpharmacological treatments including cognitive training using animal models for Alzheimer’s disease and related dementias are emerging. Dare et al. recently reported that combined physical and cognitive exercise reversed recognition memory deficits for a prolonged period and restored the acetylcholinesterase activity altered by amyloid-β neurotoxicity in the hippocampus in rats [85]. Similarly, multicomponent training, including aerobic and anaerobic physical exercise and cognitive exercise, was shown to prevent memory deficits related to amyloid-β neurotoxicity in the hippocampus in rats by Soares et al. [86]. The independent effect of cognitive training is reported to improve performance in memory tasks in ischemia-received rats [87]. These consistent findings in animal models of Alzheimer’s disease encourage the application of preoperative cognitive training in animal models for PND. These preclinical animal studies will facilitate understanding of the mechanisms of brain plasticity and guide future clinical investigations.

## 5. Conclusions

The current literature supports the benefits of cognitive training in both non-surgical and presurgical elderly patients. However, the evidence in presurgical patients is limited to the effect of the reduction of POD [74]. It is thus critical to further investigate whether preoperative cognitive training could decrease the incidence of PND or improve the trajectory of cognitive decline in elderly surgical patients. In addition, there are a multitude of questions that need to be answered in order to establish effective cognitive training programs. First, patients with preexisting cognitive impairment are mostly excluded from cognitive training studies, and thus it is not known whether preoperative cognitive training can improve postoperative outcomes in these patients. As the studies on cognitive training in older non-surgical patients with cognitive impairment or dementia show promising results [66,68], patients with cognitive impairment, neurological disease, or psychiatric illness should be involved in studies of presurgical cognitive training. These vulnerable patients may benefit most from cognitive training. Nevertheless, cognitive screening must be incorporated in presurgical assessments for all elderly surgical patients to assess their baseline cognitive functions, so that undiagnosed cognitive impairment may be identified. The next step will be to investigate the types of cognitive training and the dosages that are most beneficial for patients with preexisting cognitive impairment. Although all elderly patients, including those with normal and abnormal cognitive screening, may benefit from preoperative cognitive training, different types and dosages of training may be required. Demonstrating the prevalence of cognitive impairment in older presurgical patients through widely implemented screening can also assist in understanding the scope of this problem and facilitate the development of effective cognitive training programs [32].

Secondly, a new or modified preoperative cognitive screening that can be performed by phone or video call may be helpful in order to implement cognitive training in limited presurgical time. Presurgical visits to in-person clinics have been declining, a trend which is more evident since the COVID-19 pandemic. Remote cognitive screening will help provide screening as soon as surgery is scheduled and start cognitive training programs without delay. Whether cognitive training should be continued through the postoperative recovery period will be guided by ongoing studies. Thirdly, clinical and preclinical studies suggest that combining preoperative cognitive training and other interventions likely provides increased benefits for cognitive outcomes. Exercise is one of the most studied interventions and is suggested to improve postoperative cognitive functions when provided alongside cognitive training [86]. Interventions based on the possible pathophysiology of PND, such as the application of anti-inflammatory agents for the prevention of neuroinflammation, will likely be effective when combined with cognitive training [88,89]. Any modification of anesthetic agents may need to be tested together with cognitive training, as the anesthetics can be a predisposing and precipitating factor of PND due to their disruption of the neurotransmitter balance in the brain [21,22]. Lastly, preclinical research, specifically brain plasticity and the pathophysiology of PND, is essential for establishing the most effective cognitive training in elderly surgical patients.

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
