# Peer review of "Does Preoperative Cognitive Optimization Improve Postoperative Outcomes in the Elderly?"

_jcm, 2022, doi:10.3390/jcm11020445_

Round 1
Reviewer 1 Report
The concept of preoperative cognitive optimization is new and of high interest. The prehabilitation strategy in surgical patients is largely applied and the preoperative cognitive improvement is under scrutiny. Therefore the paper is of high interest and highlights the existing data in this area. The paper is well documented (comprehensive significant bibliography) and well organized in subchapters, which logically approach experimental and clinical data in surgical and non-surgical patients. The paper is well written with a good flow of ideas, is easy to read and understand.Author Response
Thank you for taking time and reviewing this manuscript. I deeply appreciate your expertise and comments. The revised manuscript will be submitted to the JCM submission site. Thank you again.
Reviewer 2 Report
This manuscript reviews perioperative neurocognitive dysfunction (PND). The author discusses the proposed causes of PND as well as recent studies that demonstrate the advantages of preoperative cognitive training for elderly patients.
I found this review complete with appropriate references. There are only a few minor edits:
- Please assure that the font is uniform throughout the text. There are several points (e.g. line 87-88) where the font is changed.
- The link listed on line 109 should be listed in the references. However, this is minor and could stand in the text with a uniform font as per #1.
- The abstract (line 14 and 16) uses the editorial "we," but this is a single author manuscript. The author should easily be able to revise these small edits.
Author Response
I truly appreciate the Reviewer’s expertise and comments provided for this manuscript. I have done my best to correct the shortcomings identified by the Reviewer.
- Please assure that the font is uniform throughout the text: I have corrected these fonts that are changed.
- The link listed on line 109 should be listed in the references: I have listed this citation in the references (Gauthier, S.; Rosa-Neto, P.; Morais, J.A.; Webster, C. World Alzheimer Report 2021 Journey through the diagnosis of dementia; Alzheimer’s Disease International: The International Federation of Alzheimer’s Disease and Related Disorders Societies, Inc.: London, UK, September 2021). I also listed another report citation in text, National Health Statistics Reports (line 34-35), in the reference. I thank for the reviewer for bringing up this issue.
- The abstract (line 14 and 16) uses the editorial "we," but this is a single author manuscript: I have corrected these in the abstract.
Reviewer 3 Report
The review article by Ishizawa summarizes the current evidence for preoperative cognitive training to prevent or reduce perioperative neurocognitive disorder. As there is a lack of studies assessing the effect of preoperative cognitive exercises in surgical patients, the article also includes literature on the effects of cognitive training in non-surgical patients.
Overall, the article is nicely written, and the topic is of interest. However, although being a narrative review, the author should shortly outline how the references were chosen and whether there was a specific (Pubmed) search with specific terms. Furthermore, the author should screen clinicaltrials.gov for ongoing studies on this topic, as not all trial protocols are published in journals in advance.
Please find one additional minor point below.
P2L84-87: I don’t understand the controversy between the two subsentences. Probably “although” isn’t the right word to start with.
Author Response
I truly appreciate the Reviewer’s expertise and comments provided for this manuscript. I have done my best to correct the shortcomings identified by the reviewer.
The author should shortly outline how the references were chosen and whether there was a specific (PubMed) search with specific terms. Furthermore, the author should screen clinicaltrials.gov for ongoing studies on this topic…: Thank you for raising this point. I have added the sentences to indicate how the literature search was performed in the introduction. I have performed the search in PubMed and ClinicalTrials.gov using the listed keywords. At present for the resubmission, the manuscript seems updated in this regard and no new studies or clinical trials are added in the resubmission.
P2L84-87: I don’t understand the controversy between the two subsentences. Probably “although” isn’t….: Thank you for pointing out this issue. I have revised in the resubmission.